# A Novel Integrated Path Planning and Mode Decision Algorithm for Wheel–Leg Vehicles in Unstructured Environment

**DOI:** 10.3390/s25092888

**Published:** 2025-05-03

**Authors:** Kui Wang, Xitao Wu, Shaoyang Shi, Mingfan Xu, Yifei Han, Zhewei Zhu, Yechen Qin

**Affiliations:** School of Mechanical Engineering, Beijing Institute of Technology, Beijing 100081, China; 3120230321@bit.edu.cn (K.W.); 7520230236@bit.edu.cn (X.W.); shishaoyang@bit.edu.cn (S.S.); 3220225075@bit.edu.cn (M.X.); 3120220334@bit.edu.cn (Y.H.); zhuzhewei@bit.edu.cn (Z.Z.)

**Keywords:** wheel–leg vehicles, Markov decision, reinforcement learning, mode decision

## Abstract

Human exploration and rescue in unstructured environments including hill terrain and depression terrain are fraught with danger and difficulty, making autonomous vehicles a promising alternative in these areas. In flat terrain, traditional wheeled vehicles demonstrate excellent maneuverability; however, their passability is limited in unstructured terrains due to the constraints of the chassis and drivetrain. Considering the high passability and exploration efficiency, wheel–leg vehicles have garnered increasing attention in recent years. In the automation process of wheel–leg vehicles, planning and mode decisions are crucial components. However, current path planning and mode decision algorithms are mostly designed for wheeled vehicles and cannot determine when to adopt which mode, thus limiting the full exploitation of the multimodal advantages of wheel–leg vehicles. To address this issue, this paper proposes an integrated path planning and mode decision algorithm (IPP-MD) for wheel–leg vehicles in unstructured environments, modeling the mode decision problem using a Markov Decision Process (MDP). The state space, action space, and reward function are innovatively designed to dynamically determine the most suitable mode of progression, fully utilizing the potential of wheel–leg vehicles in autonomous movement. The simulation results show that the proposed method demonstrates significant advantages in terms of fewer mode-switching occurrences compared to existing methods.

## 1. Introduction

Autonomous vehicles serve as effective transportation tools for exploration and rescue in complex and unknown environments, capable of operating independently without direct human intervention or supervision [1,2,3,4,5,6,7,8,9]. The authors of [1] proposed a radar-based fusion perception algorithm that provides perceptual information for autonomous vehicle motion. The researchers of [2] introduced a vehicle communication technology that enhances the safety and efficiency of autonomous vehicles. Also, ref. [3] developed a shock-absorbing-aware trajectory planning algorithm for autonomous vehicles, improving their navigation performance on unstructured roads. The authors of [4] proposed a more comprehensive trajectory planning method that considers a wider range of influencing factors. Furthermore, ref. [5] elucidated the current state of vehicle and pedestrian target detection, laying the foundation for navigation systems. The researchers of [6] integrated real-time perception information with trajectory planning technology, achieving autonomous navigation functionality for vehicles. On flat terrain, traditional wheeled vehicles exhibit excellent maneuverability, characterized by rapid movement and high efficiency [10,11]. The authors of [10] reviewed the current state of vehicle technologies and future trends, highlighting the superior performance of traditional wheeled vehicles on flat terrains. Moreover, ref. [11] discusses the widespread applications of conventional wheeled vehicles and proposes a novel speed estimation algorithm. However, when faced with unstructured environments, due to the limitations of the chassis and drivetrain, their maneuverability inevitably declines [12,13]. The authors of [12,13] identify the limitations of conventional wheeled vehicles and traditional navigation algorithms in lunar environments, and subsequently propose novel navigation algorithms tailored for extraterrestrial terrain. Beyond these applications, wheel–leg vehicles demonstrate particular utility in disaster response and rescue operations. These systems can rapidly approach target zones via wheeled locomotion before transitioning to legged mode for traversing debris piles or confined spaces.

To address the issues of poor maneuverability and timeliness of traditional vehicles on unstructured roads, wheel–leg vehicles have increasingly garnered attention in recent years [14,15,16,17,18]. For instance, ref. [14] proposes a wheel–leg locomotion configuration specifically designed for lunar environments, demonstrating superior off-road mobility compared to conventional wheeled vehicles. In [15], an adaptive control algorithm is developed that is tailored to this novel wheel–leg architecture. The authors of [16,17] introduce a Kalman filter-based state estimation method optimized for the new hybrid locomotion system. Relatedly, ref. [18] presents a model predictive control (MPC) algorithm for enhanced motion control of wheel–leg vehicles. Wheel–leg vehicles achieve both speed and efficiency as well as the ability to traverse complex terrains by integrating leg mechanisms with traditional wheeled structures. Electrification technology further enhances the performance and sustainability of these vehicles, with electric drive systems improving energy efficiency and reducing maintenance costs and environmental impacts. This design supports wheeled, legged, and hybrid modes of operation, significantly enhancing maneuverability and passability, thereby enabling the execution of various autonomous tasks, including unmanned driving. To achieve autonomous navigation in unstructured environments, wheel–leg vehicles require perception, planning and decision-making, and control [19,20]. Among these, planning and decision-making, especially mode decisions, represent a crucial component of autonomous task completion. Currently, research on path planning primarily focuses on 3-dimensional path planning for drones, 2.5-dimensional path planning for quadruped robots, and path planning for purely wheeled vehicles.

Drone path planning technologies can be divided into two main categories: sampling-based methods [21,22,23,24] and artificial intelligence-based methods [25,26,27]. Typical algorithms include RRT [21,22], A* [23], RRT* [24], and others. Artificial intelligence-based methods primarily include heuristic search, brute force search, and local search techniques. Park et al. [25] proposed the DroneNetX framework for relay deployment and connectivity exploration in ad-hoc networks. Subsequently, brute force search techniques were developed, with Sharma et al. [26] constructing a depth-first search algorithm for aerial mobile robots. However, this method is susceptible to interruptions during the search process, making it difficult to find efficient paths, thus leading to the emergence of local search techniques. Huang et al. [27] introduced a local search technique to ensure the shortest path in drone path planning. But it lacks interaction with the environment and cannot be applied in wheel–leg vehicles.

For quadruped robots, 2.5D path planning algorithms (which refer to planning in a two-dimensional terrain augmented with height information to represent elevation and 3D geometric features) are primarily based on the geometric environment. Wermelinger et al. [28] used various terrain representations for planning. Most planning methods compute a geometric traversability value [29] for each terrain patch to select the mode that should be adopted during the robot’s motion. Purely geometric methods are insufficient for navigation in natural outdoor environments, relying on semantic information, thus facing the same issues as traditional geometric methods [30,31]. Purely wheeled robot path planning algorithms mainly include sampling-based algorithms, artificial potential field algorithms, and graph search algorithms. The Rapidly exploring Random Tree (RRT) algorithm [32] is a typical sampling-based method, where the RRT planner samples path points within the search space. The artificial potential field (APF) algorithm [33] constructs repulsive potential fields around obstacles and attractive potential fields at the target location. In graph search algorithms [34,35], the A* algorithm performs heuristic searches and can easily find the optimal path.

However, current path planning algorithms are primarily designed for single-mode terrestrial vehicles, lacking mode decision-making in the path, and are not suitable for multi-mode wheel–leg vehicles. If the 2.5D navigation algorithm designed for legged robots is employed, the wheel–legged vehicle must operate exclusively in legged mode throughout its trajectory. While this configuration enhances obstacle-crossing capability compared to conventional wheeled vehicles, it significantly compromises locomotion efficiency on flat terrains. Conversely, adopting navigation algorithms developed for traditional wheeled vehicles fails to utilize the obstacle negotiation advantages offered by the legged modality of hybrid wheel–leg systems. Therefore, to maximize the efficiency and obstacle avoidance capability of wheel–leg vehicles, it is necessary to integrate a mode decision module into traditional 2.5D path planning algorithms to dynamically determine the most appropriate driving mode. Currently, there are no directly applicable modal decision-making frameworks available; however, decision models from other domains may serve as potential references [36,37].

In response to the issues raised, this paper first proposes an integrated path planning and mode decision method (IPP-MD) for wheel–leg vehicles. It models the mode decision problem of wheel–leg vehicles using a Markov Decision Process (MPD), innovatively designing the state space, action space, and reward function. The main contributions of this paper are summarized as follows:A navigation framework specifically designed for wheel–leg vehicles is proposed, combining a 2.5D path planning algorithm with a mode decision algorithm. Initially, the overall route of the vehicle is determined using the 2.5D path planning algorithm. Subsequently, intelligent switching of travel modes is achieved by incorporating the mode decision algorithm.A reinforcement learning–based mode decision method for wheel–leg vehicles is proposed, which reduces the model’s computational power requirements and addresses the lack of mode selection in path planning.

The paper is organized as follows: Section 2 details the proposed IPP-MD method. Section 3 presents experimental validation, and Section 4 concludes with future research directions.

## 2. Integrated Path Planning and Mode Decision Method

Figure 1 illustrates the proposed framework, which consists of a dual-layer architecture incorporating both 2.5D path planning and motion mode decision-making layers. The 2.5D path planning layer utilizes elevation maps and performance thresholds of wheel–leg vehicles to calculate their driving routes. This level of path planning transcends traditional two-dimensional path planning by integrating the efficiencies of wheeled modes and the obstacle-negotiating capabilities of legged modes. Furthermore, this path planning approach is an enhancement of the classical A* algorithm, incorporating elevation gradients into its cost function.

In the mode decision-making layer, mode selection is based on the trajectories derived from the 2.5D path planning layer, enabling determination of the appropriate driving mode for the vehicle hased on the path. The mode decision problem is modeled using a Markov process, innovatively designed with state spaces, action spaces, and a reward function. The state space includes the immediate vicinity of the current time step to allow the strategy to access more comprehensive state information. This state space is dynamically updated as time progresses, reducing redundancy in the information gathered and the computational load of the strategy. The reward function considers both safety and efficiency, aiming to maximize traffic efficiency under the premise of ensuring safety through mode decisions.

### 2.1. 2.5D Path Planning

The classical A* algorithm is a graph search algorithm typically used for global path planning. It can achieve the shortest path for a robot to move between two prescribed positions in a known static environment.

To enable the robot to autonomously avoid areas that are impassable due to slopes, steps, and uneven terrain, this paper enhances the A* algorithm in the 2.5D path planning method by adding a cost function related to terrain elevation, which incorporates traversability costs as follows:(1)$$F\left(n\right)=G_{s}\left(n\right)+H_{g}\left(n\right)$$
where *n* is the index of the current grid reached by the planned path, $G_{s}\left(n\right)$ is the minimum accumulated cost for the robot to move from its start grid *s* to the current grid *n*, and $H_{g}\left(n\right)$ is the estimated minimum cost for the robot to move from *n* to the goal grid *g*.

The cost function F(n) is applied to the A* algorithm for the global path planning in a 3D uneven terrain. According to (Equation 1), the accumulated travel cost $G_{s}\left(n\right)$ considering the influence of the terrain’s slope, step, and unevenness is(2)$$G_{s}\left(n\right)=L_{sn}+w\cdot T_{sn}$$
where $L_{sn}=\sum_{k=1}^{n-1}l_{k,k+1}$ is the accumulated Euclidean distance for the robot to travel from the start grid *s* to the current grid *n*, and $T_{sn}=\sum_{k=1}^{n-1}T_{k,k+1}$ is the accumulated traversability cost for the robot to travel between the two grids. The weighting parameter *w* balances the trade-off between path length and terrain traversability cost. A higher value of *w* biases the algorithm toward safer paths with enhanced traversability, while a lower value prioritizes shorter yet potentially more challenging routes.

The traversability cost between the *k*-th and (*k* + 1)-th grid cells is computed as follows:(3)$$T_{k,k+1}=w_{1}T_{slope}+w_{2}T_{step}+w_{3}T_{unevenness}$$
where $T_{slope}$, $T_{step}$, and $T_{unevenness}$ represent the cost of slope, elevation difference, and roughness between the *k* and k+1 grid cells, respectively, while $w_{1}$, $w_{2}$, and $w_{3}$ denote their corresponding weighting parameters.

To guarantee that the A* algorithm is optimal, the heuristic function $H_{g}\left(n\right)$ can be set as the Euclidean distance $l_{n,g}$ between *n* and *g*.

### 2.2. Mode Decision-Making

The mode decision algorithm aims to determine the appropriate mode for a vehicle, based on the elevation changes in a 2.5D path and the performance thresholds of wheel–leg vehicles. Given the future path of the vehicle, elevation map data, and basic performance parameters, it is required to ascertain the mode to be used by the vehicle as it passes through each path point. Figure 2 presents the architecture of the mode decision-making algorithm, which consists of an environmental interaction sampling module and a policy evaluation optimization module.

The task of the environmental interaction sampling module is to obtain the updated parameters of the policy neural network and reward value neural network, and then use these parameters to sample experience data from a complex environment. This module defines the problem and innovatively designs the state space, action space, and reward function. The state space is selected within a certain range of the current time step to ensure that the policy can acquire more comprehensive state information. Additionally, the time step is rolled forward to reduce redundant information gathered by the policy, thereby lowering its computational demand. The reward function takes into account the integrated performance of safety and efficiency, aiming to maximize traffic efficiency through mode decisions while ensuring safety. Ultimately, the mode decision-making problem of the wheel–leg vehicle is modeled as an MDP, which generates discrete time-series path data, including states, actions, and reward values. These data serve as the foundation for optimizing the policy neural network and reward value neural network.

The policy evaluation optimization module works in tandem with the environmental interaction sampling module, using the data collected by the sampling module to update the policy neural network and reward value neural network. It then synchronizes the updated parameters with the sampling module for the next round of sampling and optimization, continuing the process until the desired mode decision performance is achieved. This module uses Proximal Policy Optimization (PPO) to solve the Markov decision problem, employing a new objective function with clipped probability ratios, which results in a pessimistic estimation of policy performance.

#### 2.2.1. Problem Definition

The mode decision algorithm aims to determine the appropriate mode for a vehicle based on the path, based on the elevation changes in a 2.5D path and the performance thresholds of wheel–leg vehicles. Given the future path of the vehicle, elevation map data, and basic performance parameters, it is required to ascertain the mode to be used by the vehicle as it passes through each path point.

#### 2.2.2. Markov Modeling

This section converts the intelligent wheel–leg vehicle mode decision problem into a Markov Decision Problem (MDP) by defining the fundamental elements of Markov decision-making, including the global state space, action space, and reward function. This process generates discrete-time series path data, including states, actions, and reward values, which are utilized to optimize both the policy neural network and the reward neural network. The Markov decision model is defined as a tuple $\left(S,A,P,R,\rho_{0},\gamma \right)$, where the following hold:*S* represents the global state space;*A* represents the action space;P:S×A×S→[0,1] denotes the state transition probability function from state $s_{t}\in S$ and action space $a_{t}\in A$ to the next state $s_{t+1}\in S$, where *t* represents the current time step;R:S×A→R represents the joint reward function;$\rho_{0}:S\to \left[0,1\right]$ indicates the initial state distribution of the policy;γ:S→[0,1] is the discount factor.

According to the Markov decision model, the agent interacts with the environment in discrete time steps. At each time step *t*, the agent receives all information of the current state $s_{t}\in S$ from the environment and executes an action $a_{t}\in A$. Once the action is executed, a reward $R(s_{t},a_{t})$ is generated based on the reward function, and subsequently, the environment transitions to a new state $s_{t+1}∼P\left(s_{t+1}|s_{t},a_{t}\right)$. The goal of the reinforcement learning algorithm is to enable the agent to learn a policy π, based on the sample path data obtained from interactions with the environment, that maximizes the expected reward return.

#### 2.2.3. Establishment of State Space and Action Space

The global state space is used to describe the observational information gathered during the interaction between the agent and the environment, which is crucial for reinforcement learning. For the mode decision system based on the intelligent wheel–leg vehicle, the ability to effectively extract and reconstruct observational information from the wheel–leg vehicle’s driving environment determines its capability to accurately output appropriate mode decisions in challenging environments. To reduce redundant information acquired by the policy and make rational decisions, the state space considers the flow characteristics of time and the obstacle-crossing performance of the wheel–leg platform. The structural composition of the state space is presented in Figure 3, comprising three key elements: vehicle parameters, elevation map, and the path information of the vehicle. The state space is defined as S=(ξ,z,τ). Here, ξ represents the basic parameters of the wheel–leg vehicle; *z* represents the elevation map information; and τ represents the path information of the vehicle. By combining information (ξ,z,τ), the surrounding environment of the vehicle can be determined. This state space adapts to challenging scenes dynamically and effectively characterizes the obstacle-crossing performance information of the vehicle, the elevation information of the surrounding environment, and the platform’s path information within a certain time step, laying a solid foundation for the policy to make appropriate mode decisions.

The action space *A* mapped from the state space by the policy neural network is the mode command that the wheel–leg vehicle should use. The algorithm controls the anticipated mode choice of the vehicle for the next time step, facilitating coordinated traversal of the vehicle.

#### 2.2.4. Design of the Reward Function

The reward function is a quantified feedback from the environment to the agent system, reflecting the effectiveness of the action space mapped by the policy neural network. The reward function is essential for guiding the agent to continually learn with the objective of maximizing the reward value. The design of these functions critically influences the incremental performance and convergence speed of reinforcement learning algorithms. In this study, the reward function has been comprehensively and finely designed for a mode decision system based on an intelligent wheel–leg vehicle, achieving precise and robust mode decisions.

In reinforcement learning methods, the reward function plays a crucial role in guiding the agent toward achieving predefined objectives. However, these methods often fail to account for subtle variations present in complex real-world situations. This oversight may inadvertently encourage the agent to exhibit unexpected behaviors, prefer suboptimal solutions, or even compromise the system’s integrity to maximize specified rewards. To address these challenges and introduce more detailed behaviors or constraints, the concept of reward regularization has been introduced. By incorporating additional adjustments or penalties into the original reward signals, this study aims to accelerate the agent’s learning speed and simplify training. As a valuable extension of traditional influence regularization techniques, this comprehensive policy more thoroughly captures the performance and behavioral characteristics of the control system, allowing for more precise control over its actions. Thus, this research considers separate reward subcomponents triggered by specific events: mode decision completion event $r_{s}$, accident occurrence event $r_{d}$, frequent mode switching event $r_{c}$, timely mode switching event $r_{a}$, and obstacle performance violation event $r_{v}$. The first four are sparse cost assessment items triggered by specific events, and the last one is a dense reward evaluation item triggered at each time step. These reward subcomponents comprise the complete reward function, guiding the policy to learn quickly and effectively to achieve precise and robust strategies.

This study defines the mode decision completion event as(4)$$r_{s}=\delta_{s}\sum_{t=1}^{\infty }\epsilon_{s}^{t}$$
where the Boolean variable $\epsilon_{s}^{t}$ indicates whether the wheel–leg vehicle has successfully completed the predetermined path at time step *t*. If successfully reaching the destination, $\epsilon_{s}^{t}$ is set to 1; otherwise, it is 0. $\delta_{s}$ represents the discount factor for the mode decision completion event. The accident occurrence event is defined as(5)$$r_{d}=-\delta_{d}\sum_{t=1}^{\infty }\epsilon_{d}^{t}$$
where the Boolean variable $\epsilon_{d}^{t}$ indicates whether the wheel–leg vehicle has experienced a rollover, collision, or other movement-related accidents at time step *t*. If a serious safety accident occurs during exploration, $\epsilon_{s}^{t}$ is set to 1; otherwise, it is 0, and then a new round of exploration begins. $\delta_{s}$ represents the discount factor for the mode decision completion event. The frequent mode switching event is defined as(6)$$r_{c}=-\delta_{c}\sum_{t=1}^{\infty }\left[clip\left(\sum_{i=t-j}^{t+j}\epsilon_{c}^{i}-1,0\right)\right]$$
where the Boolean variable $\epsilon_{c}^{i}$ indicates whether the wheel–leg vehicle has undergone a mode switch at time step *i*; if a mode switch occurs, it is set to 1—otherwise, it is 0. *j* represents the visibility length for determining frequent mode switching events. In practical operations, frequent switching not only wastes significant time but also is not energy-efficient; thus, such behavior should be minimized. The latter part of the formula signifies that a single mode switch within a certain viewing range is considered normal behavior; more than one mode switch constitutes a frequent mode switching event, with a reward penalty based on the number of switches. $\delta_{c}$ represents the discount factor for the frequent mode switching event.

The timely mode switching event is defined as follows:(7)$$r_{a}=-\delta_{a}\sum_{t=1}^{\infty }\epsilon_{a}^{t}$$
where the Boolean variable $\epsilon_{a}^{t}$ indicates whether the wheel–leg vehicle has made a mode switch at an appropriate location at time step *t*. The platform needs to ensure a relatively unobstructed environment within a certain range during mode switching; otherwise, there might be a risk of mode switching failure. If there are no obstacles near the mode switching location, $\epsilon_{a}^{t}$ is set to 0; otherwise, it is set to 1.

The obstacle performance violation event is considered a dense reward evaluation item and is defined as follows:(8)$$r_{v}=\sum_{t=1}^{\infty }\delta_{v}\epsilon_{v}^{t}$$

If $h_{t}<h_{w}\;\mathrm{and}\;a_{t}=1\;\mathrm{or}\;h_{t}>h_{w}\;\mathrm{and}\;a_{t}=0$, $\epsilon_{v}^{t}=1$; otherwise, $\epsilon_{v}^{t}=0$. Here, $h_{t}=\frac{Z\left(x_{t+1},y_{t+1}\right)-Z\left(x_{t-1},y_{t-1}\right)}{2\Delta d}$ represents the obstacle performance based on the terrain height gradient at time step *t*, and Δd represents the distance between two adjacent path points. $\delta_{v}$ represents the discount factor for the obstacle performance violation event. In summary, the reward function includes five subcomponents: mode decision completion events $r_{s}$, accident occurrence events $r_{d}$, frequent mode switching events $r_{c}$, timely mode switching events $r_{a}$, and obstacle performance violation events $r_{v}$:(9)$$R=r_{s}+r_{d}+r_{c}+r_{a}+r_{v}$$

This reward element achieves a balanced optimization between safety and efficiency in the mode decision-making algorithm. Appropriate mode selection is critical for ensuring both operational safety and travel efficiency. Specifically, the wheeled mode proves optimal for enhanced traversal speed on relatively flat terrain, while the legged mode ensures safe navigation across steep or complex topographies.

The reward function gives quantified feedback from the environment to the agent system, reflecting the effectiveness of the action space mapped by the policy neural network. The reward function is essential for guiding the agent to continually learn with the objective of maximizing the reward value. The design of these functions critically influences the incremental performance and convergence speed of reinforcement learning algorithms. In this study, the reward function has been comprehensively and finely designed for a mode decision system based on an intelligent wheel–leg vehicle, achieving precise and robust mode decisions.

#### 2.2.5. The Policy Update Process

Figure 4 illustrates the update process of the policy neural network. The process in mode decision methods based on proximal policy optimization aims to continually update policies to maximize expected rewards by utilizing trajectories $D_{t}=\left(s_{t},a_{t},R_{t},s_{t+1}\right)$ collected through the agent–environment interaction sampling module. The object function is(10)$$J\left(\pi \right)=E_{s_{0}∼\rho_{0},a∼\pi ,s_{1:\infty }∼P}\left[\sum_{t=0}^{\infty }\gamma^{t}R\left(s_{t},a_{t}\right)\right]$$

The state–action value function $Q^{\pi }\left(s,a\right)$ under policy π represents the reward obtained by taking action *a* in state *s*, while the state value function $V^{\pi }\left(s\right)$ intuitively represents the quality of state *s*. They are defined as follows:$$Q^{\pi }\left(s,a\right)=E_{s_{1:\infty }∼P,a_{0:\infty }∼\pi }\left[\sum_{t=0}^{\infty }\gamma^{t}R\left(s_{t},a_{t}\right)\right.$$$$\left.|s_{0}=s,a_{0}=a\right]$$(11)$$V^{\pi }\left(s\right)=E_{a∼\pi }\left[Q^{\pi }\left(s,a\right)\right]$$

On this basis, the concept of the advantage function $A^{\pi }\left(s,a\right)$ is established, which represents the advantage of taking action *a* in state *s* over the average reward of any action, defined as follows:(12)$$A^{\pi }\left(s,a\right)=Q^{\pi }\left(s,a\right)-V^{\pi }\left(s\right)$$

The following identity indicates that the expected reward of the new policy $\pi_{\theta }$ is greater than that of the old policy $\pi_{\theta_{\mathrm{old}}}$ and accumulates over time steps:$$J\left(\pi_{\theta }\right)=J\left(\pi_{\theta_{\mathrm{old}}}\right)+$$(13)$$E_{s_{0}∼\rho_{0},a∼\pi_{\theta },s_{1:\infty }∼P}\left[\sum_{t=0}^{\infty }\gamma^{t}A^{\pi_{\theta_{\mathrm{old}}}}\left(s_{t},a_{t}\right)\right]$$

Let $\rho_{\pi }$ denote the discounted visitation frequencies:(14)$$\rho_{\pi }\left(s\right)=\sum_{t=0}^{\infty }\gamma^{t}P\left(s_{t}=s\right)$$

Transform Equation (Equation 13) from a sum over time steps to a sum over states:$$J\left(\pi_{\theta }\right)=J\left(\pi_{\theta_{\mathrm{old}}}\right)+\sum_{t=0}^{\infty }\sum_{s}P\left(s_{t}=s|\pi_{\theta }\right)$$$$\sum_{a}\pi_{\theta }\left(a|s\right)\gamma^{t}A^{\pi_{\theta_{\mathrm{old}}}}\left(s,a\right)$$(15)$$=J\left(\pi_{\theta_{\mathrm{old}}}\right)+\sum_{s}\rho_{\pi_{\theta }}\left(s\right)\sum_{a}\pi_{\theta }\left(a|s\right)A^{\pi_{\theta_{\mathrm{old}}}}\left(s,a\right)$$

This equation implies that for any policy update from $\pi_{\theta_{\mathrm{old}}}$ to $\pi_{\theta }$, there is a non-negative expected advantage, ensuring that each policy update can optimize the reward performance or remain unchanged in the case where the expected advantage is zero everywhere. It indicates that the outcome of iterative updates using the deterministic policy $\pi_{\theta }\left(s\right)=\arg\max_{a}A^{\pi }\left(s,a\right)$ results in improved policy performance if at least one state–action pair has a positive advantage value and the state visitation frequency is non-zero; otherwise, the algorithm converges to the optimal policy. However, due to the high dependency of $\rho_{\pi_{\theta }}\left(s\right)$ on $\pi_{\theta }$, the above equation is difficult to optimize directly. Hence, the following local approximation is introduced:$$L_{\pi }\left(\pi_{\theta }\right)=J\left(\pi_{\theta_{\mathrm{old}}}\right)+\sum_{s}\rho_{\pi_{\theta_{\mathrm{old}}}}\left(s\right)$$(16)$$\sum_{a}\pi_{\theta }\left(a|s\right)A^{\pi_{\theta_{\mathrm{old}}}}\left(s,a\right)$$

In the above formula, $\rho_{\pi_{\theta_{\mathrm{old}}}}$ is used in place of $\rho_{\pi_{\theta }}$, disregarding the changes in state visitation frequency due to changes in policy. The rationale is that with sufficiently small update steps, improvements in $L_{\pi }\left(\pi_{\theta }\right)$ necessarily lead to enhancements in $J(\pi_{\theta })$:(17)$$L_{\pi_{\theta_{0}}}\left(\pi_{\theta }\right)=J\left(\pi_{\theta_{0}}\right)$$(18)$$\nabla_{\theta_{0}}L_{\pi_{\theta_{0}}}\left(\pi_{\theta }\right){{|}_{\theta_{0}}=\nabla_{\theta_{0}}J\left(\pi_{\theta_{0}}\right)|}_{\theta_{0}}$$

Thus, under the condition of sufficiently small update increments, the objective function can be approximated as follows:(19)$$L_{\pi }\left(\pi_{\theta }\right)=E_{a_{t}∼\pi_{\theta_{\mathrm{old}}},s_{t}∼P}\left[\frac{\pi_{\theta }\left(s_{t},a_{t}\right)}{\pi_{\theta_{\mathrm{old}}}\left(s_{t},a_{t}\right)}A^{\pi_{\theta_{\mathrm{old}}}}\left(s_{t},a_{t}\right)\right]$$$$s.t.E_{a_{t}∼\pi_{\theta_{\mathrm{old}}},s_{t}∼P}\left[D_{KL}\left(\pi_{\theta_{\mathrm{old}}}\left(\cdot |s_{t}\right)∥\pi_{\theta }\left(\cdot |s_{t}\right)\right)\right]\leq \delta$$
where $D_{KL}\left(\pi_{\theta_{\mathrm{old}}}\left(\cdot |s_{t}\right)∥\pi_{\theta }\left(\cdot |s_{t}\right)\right)$ represents the KL-divergence gap between the actions taken by the new and old policies under the same state $s_{t}$, and δ is the KL-divergence threshold. The choice of δ is crucial; if the step size is too large, it can lead to unstable updates; if too small, it results in excessively slow updates. During the update process, the characteristics of the step size threshold may change, and a single KL-divergence threshold cannot meet the needs of the complete training process, necessitating certain modifications to the equation. Let $r_{t}\left(\theta \right)$ represent the probability ratio $\frac{\pi_{\theta }\left(s_{t},a_{t}\right)}{\pi_{\theta_{\mathrm{old}}}\left(s_{t},a_{t}\right)}$ when the policy remains unchanged, $r_{t}\left(\theta_{\mathrm{old}}\right)=1$. To limit the disparity between the new and old policies, consider penalizing policy changes when $r_{t}\left(\theta_{\mathrm{old}}\right)$ deviates from one, modifying the objective function as follows:$$L_{\pi_{\theta_{0}}}^{\prime }\left(\pi_{\theta }\right)=E_{a_{t}∼\pi_{\theta_{\mathrm{old}}},s_{t}∼P}[\min(r_{t}\left(\theta \right)A^{\pi_{\theta_{\mathrm{old}}}}\left(s_{t},a_{t}\right),$$(20)$$\mathrm{clip}\left(r_{t}\left(\theta \right),1-\epsilon ,1+\epsilon \right)A^{\pi_{\theta_{\mathrm{old}}}}\left(s_{t},a_{t}\right))]$$

Here, the hyperparameter ϵ belongs to the interval [0, 1]. The objective function of this equation consists of two parts: the first part is the original objective function term; the second part is adjusted by clipping the probability ratio, specifically $\mathrm{clip}(r_{t}\left(\theta \right),1-\epsilon ,1+\epsilon )$, which limits the probability ratio to within the interval [1−ϵ,1+ϵ]. The algorithm selects the lesser of the clipped objective function and the original objective function as the final goal. Under this strategy, when the advantage function takes a positive value and the probability ratio exceeds the upper limit, the excess will be clipped, thus limiting the difference between the new and old policies. Conversely, when the advantage function is negative and the probability ratio exceeds the lower limit, this part of the effect will not be ignored; adjustments are only made when the changes in the probability ratio are beneficial to the objective function. This ensures that strategy differences are considered only when they are advantageous for optimizing the objective. The advantage function $A^{\pi_{\theta_{\mathrm{old}}}}\left(s_{t},a_{t}\right)$ can be calculated using (Equation 21) where λ is a smoothing parameter used to reduce variance and stabilize training:(21)$$A^{\pi_{\theta_{\mathrm{old}}}}\left(s_{t},a_{t}\right)=\delta_{t}+\left(\gamma \lambda \right)\delta_{t+1}+\ldots +{\left(\gamma \lambda \right)}^{T-t}\delta_{T}$$(22)$$\delta_{t}=r_{t}+\gamma V_{\Phi }\left(s_{t+1}\right)-V_{\Phi }\left(s_{t}\right)$$
where $V_{\Phi }\left(s_{t}\right)$ is the value function approximated by a reward value neural network, with parameters Φ updated through the following equation:(23)$$L\left(\Phi \right)=E\left[{\left(V_{\Phi }\left(s_{t}\right)-R\left(s_{t}\right)\right)}^{2}\right]$$

## 3. Experiments Validation

This section describes the experiments conducted within the ROS system and visualized in the Rviz environment to evaluate the performance of the proposed method. The models and parameters used in the experimental implementation are first introduced, followed by an analysis of the obtained training and testing results. Finally, the proposed method is compared with rule-based approaches in terms of performance.

### 3.1. Experimental Setup

The hardware configuration used in the experiments consisted of an Intel i7-13700KF processor with 16 GB of RAM and an NVIDIA GeForce RTX 4060 GPU. The software environment was based on the Ubuntu 20.04 operating system, with inter-module communication facilitated by the ROS (Robot Operating System) framework. The deep reinforcement learning model was implemented using the PyTorch 1.13.0+cu116 framework. The policy neural network integrated map information, path data, and the performance characteristics of the wheel–legged vehicle to determine the locomotion mode selection at each path point. Figure 5 displays the vehicle configuration and elevation map. The training map used in the experiments includes both depressions and hilly terrains. The entire map measures 80 m in both length and width, with a resolution of 0.156 m, with depressions and hilly terrains each occupying half of the map. The IPP-MD method first trains the agent on the map, where the agent collects trajectory data through interaction with the environment and updates the policy using this trajectory data. This process is repeated until the agent achieves the desired performance. To evaluate the performance of the proposed IPP-MD method, two sets of experiments were conducted. The first experiment provides a detailed analysis of the training process of the IPP-MD method, while the second experiment deploys the trained model and tests it across various field terrains, comparing its computational time and mode-switching frequency performance with a rule-based mode decision method. Utilizing regions different from the training set allows for the validation of the algorithm’s generalization capability. The IPP-MD algorithm employs a multilayer perceptron with four hidden layers, each containing 1024 units, to approximate the policy and value functions. The learning rate decreases linearly from $10^{-3}$ to 0, and the standard deviation decays exponentially from 1 to 0.

### 3.2. Training Process Analysis

The training consists of 6000 iterations, with each iteration comprising 2000 time steps. The Adam optimizer is used for policy updates, and the score changes during training are shown in Figure 6. The figure represents the average value across five training runs for each iteration, illustrating the trend in reward changes during the training process. The shaded area indicates the value boundaries from the five training runs, reflecting the fluctuation of reward values during training.

In the early stages of learning, influenced by the terrain adaptation mode selection reward component, the policy initially prioritizes learning how to select the appropriate mode based on terrain information, in order to avoid performance waste and efficiency loss. This phase corresponds to the first 500 iterations in the curve. As learning progresses, the policy increasingly avoids frequent mode switching, as long as performance thresholds are not violated. This is because the penalty for violating performance thresholds in the reward function is greater than the penalty for frequent mode switching. This phase corresponds to iterations 500–2500 in the curve. Finally, between iterations 2500 and 6000, the curve stabilizes, resulting in a deployable mode decision model. Overall, the curve exhibits a relatively small shaded area, indicating that the proposed policy demonstrates good robustness.

### 3.3. Hill Terrain Validation

The algorithm was validated on hill terrain, and the mode decision results are shown in Figure 7. Figure 7a,b display the IPP-MDalgorithm, while Figure 7c,d present the rule-based mode decision algorithm. Figure 7a,c show the overhead views of both algorithms, and Figure 7b,d offer a clearer perspective to illustrate the variation in terrain height. The colors along the path represent the mode used at each point: green indicates the wheeled mode, and red indicates the legged mode.

From the overhead view, it can be seen that the path bypasses the highest region at the peak of the mountain, as the wheel–leg vehicle’s maximum performance cannot traverse over the peak. The path passes through the middle of the slope, where there is some elevation and gradient compared to flat terrain, but it remains within the vehicle’s performance threshold. In hill terrain, it is unrealistic to plan a path that can be traversed solely using the wheeled mode, and relying exclusively on the legged mode would waste a significant amount of time. Therefore, the mode decision algorithm for the wheel–leg vehicle is crucial in this terrain.

The main mode switching processes in the path are represented by three circles in Figure 7b,d. The blue circle represents the major mode switching segment in the uphill condition, the red circle represents the major mode switching segment at the steep terrain at the peak, and the gray circle represents the major mode switching segment in the downhill condition. The first major mode switch occurs in the blue circle, where the IPP-MDalgorithm shows that the wheel–leg vehicle switches from the wheeled mode to the legged mode before reaching the slope area, enabling the vehicle to climb the slope. In the rule-based mode decision algorithm, frequent mode switching occurs within the gray circle, resulting in significant time wastage. Moreover, performing mode switching in the middle of the slope is relatively dangerous for the wheel–leg robot. The IPP-MDalgorithm avoids this mode switching behavior through the penalty term for frequent mode switching included in the reward function, encouraging the agent to avoid such behavior during the learning process in order to maximize the reward. The second red circle represents a short and steep terrain. The rule-based mode decision algorithm uses the legged mode in this short path, but it cannot ensure that the wheel–leg vehicle can completely traverse the obstacle using the legged mode. In contrast, the IPP-MDalgorithm appropriately extends the path length in the legged mode to ensure the vehicle can fully overcome the obstacle. The third gray circle represents the downhill condition, where the vehicle needs to use the legged mode to descend. Similar to the uphill condition, the IPP-MDalgorithm maintains the legged mode throughout the descent, while the rule-based mode decision algorithm frequently switches modes during the descent, leading to time wastage and creating dangerous conditions.

Figure 8 shows a comparison of the number of mode switches and computational time between the IPP-MD algorithm and the rule-based mode decision algorithm across six experiments with different endpoints. Compared to the rule-based mode decision algorithm, the IPP-MD algorithm reduced the number of mode switches by 79.2%. Furthermore, the IPP-MD algorithm has similar computational time to the rule-based algorithm, indicating that the algorithm significantly improves mode decision performance without a significant increase in computational cost. The specific data are presented in Table 1.

### 3.4. Depression Terrain Validation

A depression terrain was selected for algorithm validation, and the mode decision-making results are shown in Figure 9. Figure 9a,b display the IPP-MDalgorithm, while Figure 9c,d show the rule-based mode decision algorithm. Figure 9a,c provide top–down views of both algorithms, and Figure 9b,d present a clearer perspective showing changes in terrain height.

From the overhead view, it can be seen that the path does not deviate significantly, as there are no terrain features in the depression that exceed the performance threshold of the wheel–leg vehicle. The path combines both wheeled and legged modes of the wheel–leg vehicle to reach the destination.

The main mode switching processes in the path are represented by two circles in Figure 9b,d. The blue circle represents the major mode switching segment in the downhill condition, and the gray circle represents the major mode switching segment in the uphill condition. The first blue ellipse represents the downhill segment in the depression terrain, where the terrain is rugged and uneven. The rule-based mode decision algorithm requires frequent mode switches to enable the wheele–leg platform to meet the obstacle-crossing performance threshold. In contrast, the IPP-MDalgorithm chooses to maintain the legged mode throughout the entire downhill segment, significantly reducing the number of mode switches while ensuring efficiency and avoiding unsafe behaviors caused by mode switching on downhill terrain. The second gray ellipse corresponds to the uphill terrain in the depression. Similar to the downhill segment, the rule-based mode decision algorithm frequently switches modes during the uphill process, whereas the IPP-MDalgorithm avoids this situation.

Figure 10 compares the IPP-MDalgorithm and the rule-based mode decision algorithm in terms of mode switching frequency and computational time across six experiments with different endpoints. Compared to the rule-based mode decision algorithm, the IPP-MD algorithm reduces the number of mode switches by 72.1% without significantly increasing computational power. The specific data are presented in Table 2.

### 3.5. Comparative Analysis of Path Planning Computational Time

To validate the computational efficiency of the proposed algorithm, a comparative analysis was conducted between the PPO-MoDeL method and the advanced pseudo-random sampling (PRS) method [38] in terms of computation time. Multiple routes were selected for testing both methods, and the experimental results are presented in Figure 11. As illustrated in the figure, the PPO-MoDeL method exhibits significantly shorter computation time, thereby demonstrating its superiority in computational efficiency. Both methods show similar fluctuations in computation time, indicating that their performance varies with route length. Nevertheless, the computational time of both methods remains within acceptable limits, ensuring real-time applicability for path planning algorithms.

### 3.6. Reward Function Component Ablation Study

To validate the effectiveness and rationality of the reward mechanism, relevant ablation experiments were conducted, with the experimental results illustrated in Figure 12. The experiments systematically eliminated each reward component individually to investigate the role of each reward term in the algorithm’s training process. As shown in the figure, the original parameter configuration achieved the highest reward score, demonstrating that the combination of the original reward function yields the best algorithmic performance. Among these components, the event completion reward $r_{s}$ had the least impact on algorithmic performance, as this term primarily accelerates training convergence rather than influencing the final performance. In contrast, the accident occurrence reward $r_{d}$ exerted the most significant influence, as this term ensures the safety of the wheel–legged vehicle, which serves as the fundamental prerequisite for successful route completion. The remaining three reward terms guided the algorithm in making appropriate modal decisions, thus exerting a moderate influence on the final performance.

### 3.7. Environmental Parameter Sensitivity Analysis

To investigate the influence of environmental parameters on algorithm performance, a sensitivity analysis was conducted. A height scaling factor was introduced to proportionally adjust the terrain elevation in the digital elevation map. As illustrated in Figure 13, a scaling factor of one maintains the original terrain elevation, values below one reduce the terrain height proportionally, and values above one amplify it. The experimental results demonstrate a positive correlation between terrain height amplification and the frequency of modality transitions. Specifically, as terrain elevation increases, the system exhibits a higher number of wheel-to-leg modality switches. This phenomenon occurs because heightened terrain variations elevate locomotion risks for the wheel–leg vehicle, triggering more frequent transitions from the wheeled to legged modality to ensure safe navigation, thereby increasing the overall modality switching count.

### 3.8. Real-World Vehicle Validation

To validate the robustness of the algorithm in real-world environments, we conducted field tests using a self-developed wheel–leg vehicle as the experimental platform. The vehicle is equipped with a distributed sensor network comprising inertial measurement units and LiDAR, with localization achieved through LiDAR odometry. The experiments were performed in unstructured off-road terrain, including uphill slopes and sandy surfaces. As shown in Figure 14, the starting point was selected in a lower-altitude area, while the destination was set in a higher-altitude region. Figure 14 illustrates the actual motion trajectory of the wheel–legged vehicle: it first traversed a relatively flat terrain, then ascended an inclined slope, and finally crossed another flat section to reach the target location.

The experimental results demonstrate that the wheel–legged vehicle can successfully navigate through unstructured off-road terrain, confirming the robustness and effectiveness of the proposed algorithm. Figure 15 presents a comparison between the reference speed and the actual achieved speed during motion. The data reveal a certain discrepancy between the two, with an average speed error of 0.05 m/s and a maximum error of 0.33 m/s. This deviation is primarily attributed to real-world factors such as sensor noise. Nevertheless, the vehicle successfully reached the target destination, fulfilling the functionality of the navigation system.

### 3.9. Results and Discussion

To validate the effectiveness and robustness of the algorithm, this study conducted extensive experiments encompassing training process analysis, hill terrain validation, depression terrain validation, comparative analysis of path planning computational time, reward function component ablation experiments, environmental parameter sensitivity analysis, and real-world vehicle validation. The hill and depression terrain validations demonstrated the method’s effectiveness in simulated environments through performance comparisons with state-of-the-art modal decision-making algorithms. The computational time comparison validated the algorithm’s superiority in real-time performance, while the reward function ablation experiments confirmed the rationality of the reward design. Environmental changes may not trigger timely algorithm adjustments, and its obstacle avoidance capability depends entirely on the completeness of pre-existing elevation data—unknown obstacles not represented in the map could potentially lead to hazardous situations. Real-world vehicle tests verified the algorithm’s applicability in physical environments. However, the proposed method exhibits inherent limitations due to its reliance on static elevation maps, rendering it incapable of adapting to highly dynamic environments.

## 4. Conclusions

This study addresses the mode selection challenge in path planning for wheel–leg vehicles operating in unstructured terrains by proposing a deep reinforcement learning-based solution. The developed algorithm integrates optimized motion mode decision-making with a 2.5-dimensional path planning framework, achieving unified efficient path generation and dynamic mode transitions in complex environments. The proposed hierarchical navigation architecture fully exploits the mobility advantages of wheel–leg vehicles across diverse terrains, overcoming the limitations of conventional path planning methods that lack dynamic mode adaptation capabilities. Research demonstrates the critical importance of multi-modal motion pattern optimization, necessitating the integration of environmental perception, path planning, and mode selection through reinforcement learning frameworks to transcend traditional wheeled vehicle navigation constraints. Experimental validation confirms the framework’s effectiveness in enabling autonomous navigation across various challenging environments while maintaining operational efficiency and safety.

However, performance discrepancies emerge between simulation and physical implementations due to unmodeled real-world factors including environmental noise, complex dynamic interactions (e.g., soil deformation and moving obstacles), and hardware latency. Future research will focus on practical deployment enhancements through improved environmental modeling, noise robustness, and dynamic compensation mechanisms. Furthermore, future research will explore the application of diverse machine learning-based approaches to wheel–leg vehicles [39], validating the applicability of multi-criteria models in their navigation tasks [40,41], thereby enhancing the autonomous navigation performance of such vehicles.

## Figures and Tables

**Figure 1 sensors-25-02888-f001:**
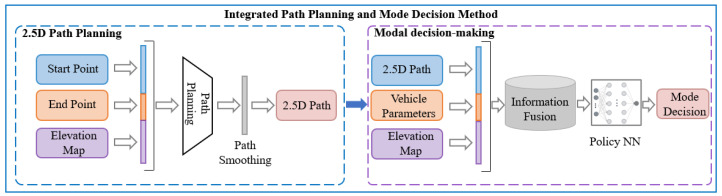
Integrated path planning and mode decision method structure diagram.

**Figure 2 sensors-25-02888-f002:**
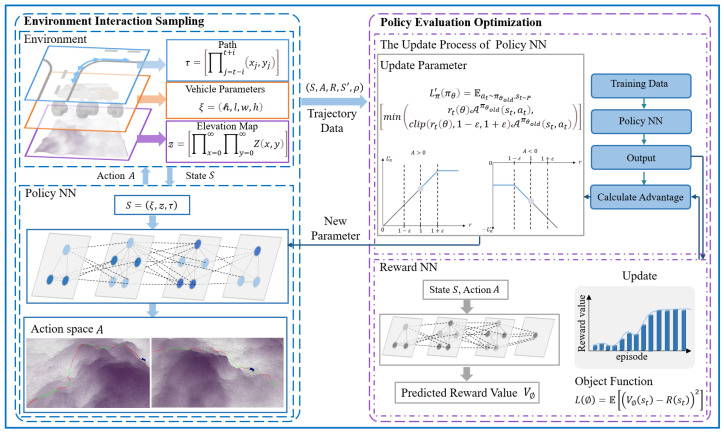
Mode decision algorithm architecture diagram. The light blue line represents the path, the orange line represents the vehicle parameters, and the purple line represents the elevation map.

**Figure 3 sensors-25-02888-f003:**
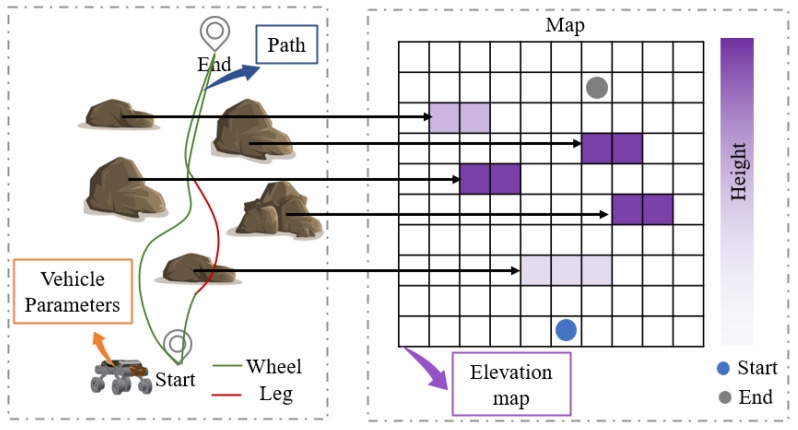
State space structure diagram.

**Figure 4 sensors-25-02888-f004:**
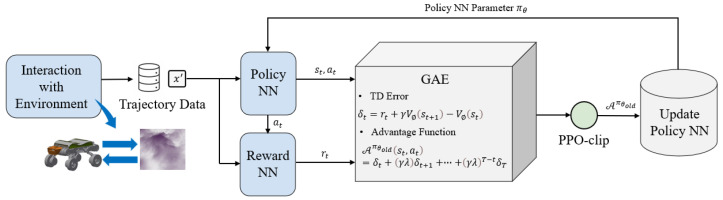
Policy neural network update process.

**Figure 5 sensors-25-02888-f005:**
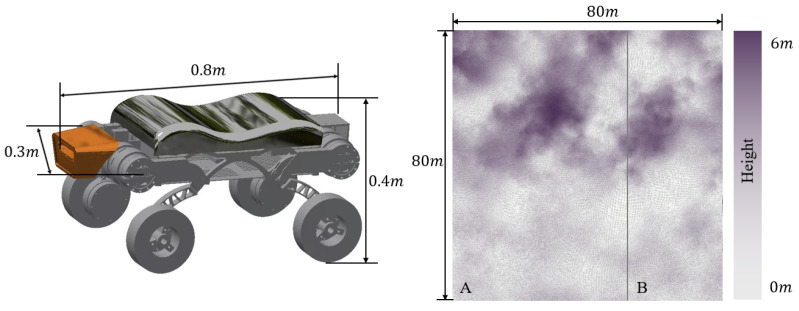
Vehicle structure and field topographic elevation map.

**Figure 6 sensors-25-02888-f006:**
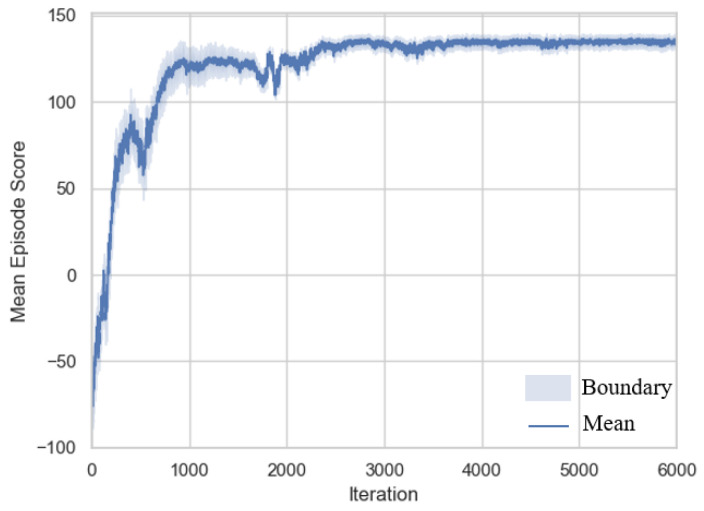
Training process reward variation.

**Figure 7 sensors-25-02888-f007:**
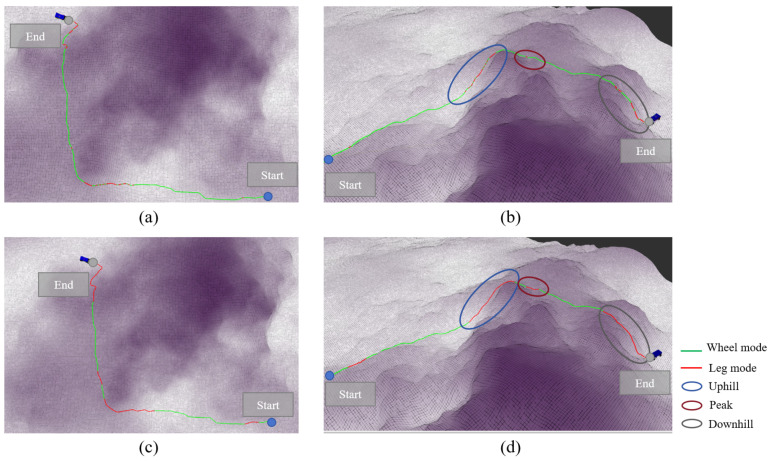
Demonstration of mode decision process in hilly terrain. (**a**) represents the top view based on rule-based methods, (**b**) represents the side view based on rule-based methods, (**c**) represents the top view based on IPP-MD methods, and (**d**) represents the side view based on IPP-MD methods.

**Figure 8 sensors-25-02888-f008:**
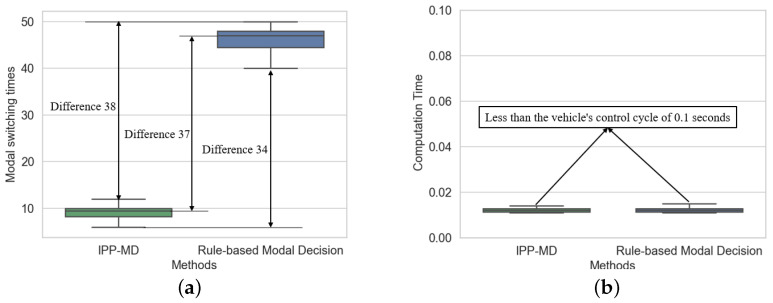
Comparison of mode switching frequency and computation time of two methods in hilly terrain. (**a**) represents the comparison of modal switching times, and (**b**) represents the comparison of calculation time.

**Figure 9 sensors-25-02888-f009:**
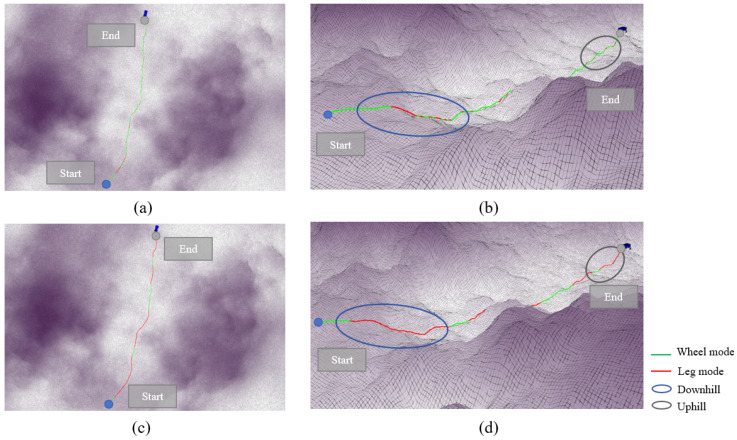
Demonstration of mode decision process in depression terrain. (**a**) represents the top view based on rule-based methods, (**b**) represents the side view based on rule-based methods, (**c**) represents the top view based on IPP-MD methods, and (**d**) represents the side view based on IPP-MD methods.

**Figure 10 sensors-25-02888-f010:**
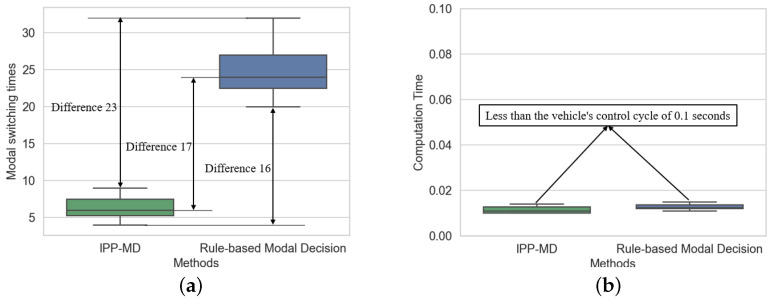
Comparison of mode switching frequency and computation time of two methods in depression terrain. (**a**) represents the comparison of modal switching times, and (**b**) represents the comparison of calculation time.

**Figure 11 sensors-25-02888-f011:**
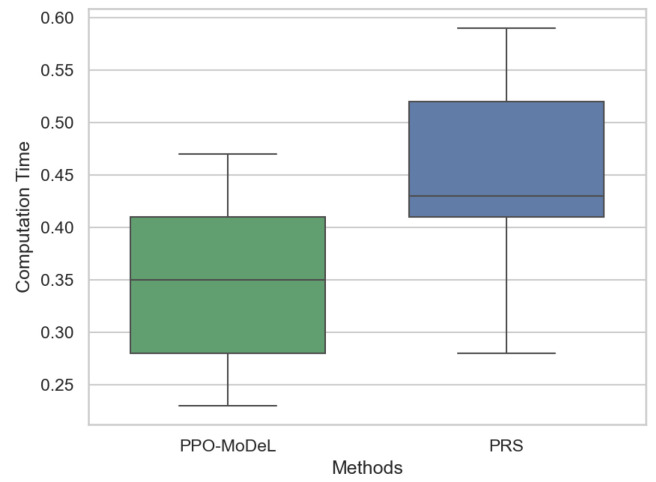
Comparative analysis of path planning computational time.

**Figure 12 sensors-25-02888-f012:**
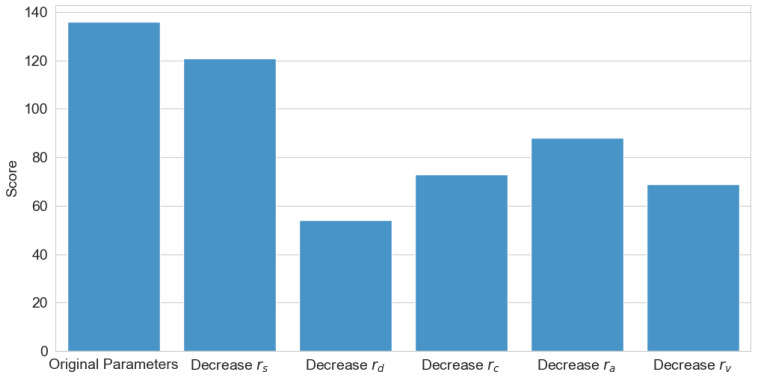
Reward function component ablation study.

**Figure 13 sensors-25-02888-f013:**
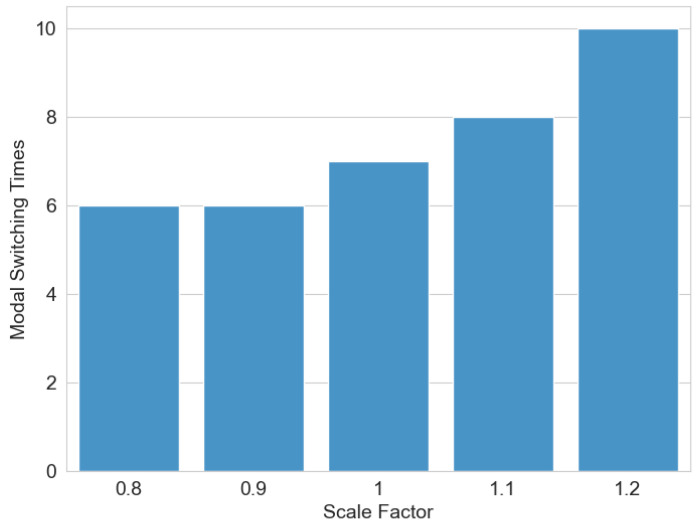
Environmental parameter sensitivity analysis.

**Figure 14 sensors-25-02888-f014:**
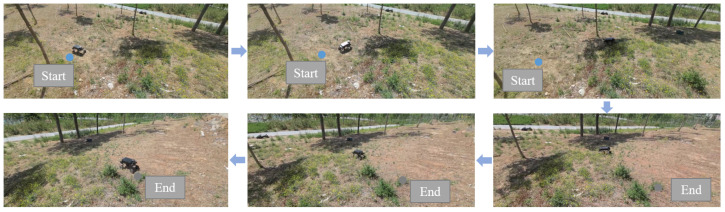
Validation in real vehicle scenarios.

**Figure 15 sensors-25-02888-f015:**
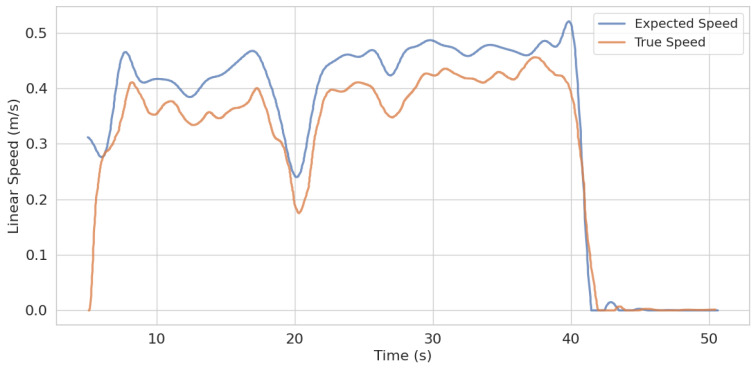
Comparison between expected speed and true speed.

**Table 1 sensors-25-02888-t001:** Performance comparison in hill terrain.

Case	Metrics	IPP-MD	Rule-Based
Case 1	Switching Times	**10**	48
	Computation Time	0.012	**0.011**
Case 2	Switching Times	**12**	50
	Computation Time	**0.011**	0.013
Case 3	Switching Times	**8**	40
	Computation Time	0.013	**0.012**
Case 4	Switching Times	**9**	44
	Computation Time	0.012	**0.011**
Case 5	Switching Times	**10**	46
	Computation Time	0.014	**0.012**
Case 6	Switching Times	**6**	48
	Computation Time	**0.011**	0.015

**Table 2 sensors-25-02888-t002:** Performance comparison in depression terrain.

Case	Metrics	IPP-MD	Rule-Based
Case 1	Switching Times	**6**	24
	Computation Time	**0.010**	0.012
Case 2	Switching Times	**8**	32
	Computation Time	0.014	**0.011**
Case 3	Switching Times	**5**	24
	Computation Time	**0.010**	0.012
Case 4	Switching Times	**4**	28
	Computation Time	**0.012**	0.013
Case 5	Switching Times	**6**	20
	Computation Time	**0.011**	0.014
Case 6	Switching Times	**9**	22
	Computation Time	**0.013**	0.015

## Data Availability

Due to the privacy of technology, the dataset is not publicly available.

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
