# Peer review of "A Novel Integrated Path Planning and Mode Decision Algorithm for Wheel–Leg Vehicles in Unstructured Environment"

_sensors, 2025, doi:10.3390/s25092888_

Round 1
Reviewer 1 Report
Comments and Suggestions for Authors
Dear Authors;
This paper presents an interesting integrated path planning and mode decision algorithm (IPP-MD) for wheel-leg vehicles. The combination of 2.5D path planning with Markov Decision Process (MDP) for mode decision is a novel approach to address the challenges of autonomous navigation in unstructured environments. However, several aspects need improvement to enhance the paper's clarity, rigor, and overall impact.
First"Major Comments":
- The introduction: - Expand the discussion of related works, especially those focusing on multi-modal locomotion or similar wheel-leg vehicles. This will better contextualize the novelty of your approach. - Provide more specific details on the limitations of existing algorithms for wheel-leg vehicles to justify the need for the proposed IPP-MD method.
- The Methodology: - Please, clearly define the state space, action space, and reward function used in the Markov Decision Process (MDP). This is critical for understanding the mode decision-making process. - Justify the choice of the weight parameter 𝑤 in Equation (2). Explain its significance and how it affects the performance of the enhanced A* algorithm. - Please, explain the criteria or thresholds used to determine traversability costs in the enhanced A* algorithm. How are these values determined or learned?- Please, explain & provide a detailed explanation of the hardware and software configurations used for the simulation experiments.
- Through the Results and Discussion, part: - please, include a discussion of the limitations of the proposed approach. For example, how does the algorithm perform in highly dynamic environments or with unforeseen obstacles?, - A comparison for the computational time of the IPP-MD algorithm with other state-of-the-art methods for path planning and mode decision must be added. - please, provide a discussion of the sensitivity of the algorithm to changes in environmental parameters or vehicle characteristics.
- all figures must be enhanced and reviwed, each figure must be discussed before the figure itself. * Second"Minor Comments": - Through the abstract, Specify the type of unstructured environments considered in this study. - Through the introduction, please Replace "address suitable issue" with "address this issue" (line 9). - Also, through the introduction, Correct "two-dimensioal" to "two-dimensional" (line 38).- please, Expand the discussion of the potential applications of wheel-leg vehicles in real-world tasks (line 32). - Through the Methodology, you must define all variables and parameters used in the equations. - Also, into the methodology, provide a more detailed explanation of how the reward function is designed to balance safety and efficiency. -Through the Results and Discussion part; Discuss the implications of the results for the design and control of wheel-leg vehicles, and please provide a more detailed discussion of the limitations of the simulation experiments.
Overall, the English language in this paper is understandable, but it requires some polishing to improve clarity and grammatical accuracy.
Author Response
We sincerely appreciate the time you have taken to review our manuscript. We have carefully addressed all your comments and revised the paper accordingly. Please refer to the PDF file for our point-by-point responses.

Reviewer 2 Report
Comments and Suggestions for Authors
The presented research answers a current scientific and technical challenge: how to ensure autonomous, safe and energy-efficient navigation of wheel-leg vehicles in conditions where traditional means of transportation lose their effectiveness.
- The main question addressed by the research is to bridge the gap between traditional path planning (mostly designed for single-mode, wheeled robots) and the multimodal capabilities of advanced wheel-leg vehicles operating in complex, real-world terrains.
- The topic is original in the field of autonomous robotics and intelligent transportation systems. The topic addresses a timely and unsolved challenge with an innovative approach in autonomous systems. The research fills a specific gap by creating a holistic intelligent system that integrates navigation and mode selection, which is critical for the real-world deployment of wheeled and legged robots on uneven, unpredictable terrain where traditional methods are ineffective.
- This study adds a transition from autonomous planning mode systems to an integrated, intelligent, terrain-aware system. This enables the development of a system with autonomous navigation in complex environments.
- Include real-world field tests with physical wheel-leg robots on natural terrains. This would validate the model’s robustness under sensor noise, hardware limitations, and dynamic environmental conditions. Conduct an ablation study by removing or changing individual reward conditions to show their importance. This would validate the reward design and explain why more effective strategies should be explored.
- The conclusion needs to be expanded in accordance with the evidence presented in the previous sections.
- The references are appropriate.
- Each figure should be preceded by a description and a reference to it.
Author Response

(The authors gave the same response as above.)

Reviewer 3 Report
Comments and Suggestions for Authors
The paper need some improvements :
-- All cited papers have to be discussed in details here " Autonomous vehicles serve as effective transportation tools for exploration and rescue in complex and unknown environments, capable of operating independently without direct human intervention or supervision [1–6]. On flat terrain, traditional wheeled vehicle exhibit excellent maneuverability, characterized by rapid movement and high efficiency [7,8]. However, when faced with unstructured environments, due to the limitations of the chassis and drivetrain, their maneuverability inevitably declines [9,10]. To address the issues of poor maneuverability and timeliness of traditional vehicles on unstructured roads,wheel-leg vehicles have increasingly garnered attention in recent years [11–15]."
--Same issue here " research on path planning primarily focuses on three-dimensional path planning for drones [19–25], 2.5-dimensional path planning for quadruped robots [26–29] and two-dimensioal path planning for purely wheeled vehicles [30–33].Drone path planning technologies can be divided into two main categories: sampling-based methods [19–22] and artificial intelligence-based methods [23–25]".
--Improve Figure 2. Mode decision algorithm architecture diagram.
--In conclusion, create a new section for future studies and describe how machine learning can be useful to estimate for this kind of complex problem, you can use this work "KAYIRAN, H. F. (2025). Machine Learning-Based Investigation of Stress in Carbon Fiber Rotating Cylinders. Knowledge and Decision Systems with Applications, 1, 1-10." and discuss how other MCDM models can be applied, such as. MARCOS Model, "AKBULUT, O. Y. (2025). Analysis of the Corporate Financial Performance Based on Grey PSI and Grey MARCOS Model in Turkish Insurance Sector. Knowledge and Decision Systems with Applications, 1, 57-69." and "Moslem, S. (2025). Evaluating commuters' travel mode choice using the Z-number extension of Parsimonious Best Worst Method. Applied Soft Computing, 173, 112918.".
--Limitations and future improvements have be highlighted clearly.
--
Author Response

(The authors gave the same response as above.)

Round 2
Reviewer 1 Report
Comments and Suggestions for Authors
Dear Authors,
Thanks for your fine work.
Author Response
Thank you for your acknowledgment. I sincerely appreciate the valuable time you have devoted to reviewing this work.

Reviewer 3 Report
Comments and Suggestions for Authors
-In the introduction, describe how other decision making models can be adopted for evaluating this kind of issue, use the related works such as, “Hussain, A., & Ali, M. (2025). A Critical Estimation of Ideological and Political Education for Sustainable Development Goals Using an Advanced Decision-Making Model Based on Intuitionistic Fuzzy Z-Numbers. International Journal of Sustainable Development Goals, 1, 23–44. Retrieved from https://ijsdg.org/index.php/ijsdg/article/view/14193”, and “Badi, I., Bouraima, M. B., Yanjun, Q., & Qingping, W. (2025). Advancing Sustainable Logistics and Transport Systems in Free Trade Zones: A Multi-Criteria Decision-Making Approach for Strategic Sustainable Development . International Journal of Sustainable Development Goals, 1, 45–55. Retrieved from https://ijsdg.org/index.php/ijsdg/article/view/14213 ”.
Author Response

(The authors gave the same response as above.)
